# Connected Driving in German-Speaking Social Media

Eugenia Rykova [1,2,*], Christine Stieben [3,*] , Olga Dostovalova [3] and Horst Wieker [3]

1 University of Applied Sciences TH Wildau, 15745 Wildau, Germany
2 University of Eastern Finland, 80100 Joensuu, Finland
3 Saarland University of Applied Sciences HTW Saar, 66117 Saarbrücken, Germany
* Correspondence: eugeniia.rykova@th-wildau.de (E.R.); christine.stieben@htwsaar.de (C.S.)

**Abstract:** Intelligent transportation systems (ITS) have been steadily becoming part of our reality. For their successful integration, studying and understanding public opinions and acceptance is important. Social media platforms offer an extensive opportunity for opinion mining. While there have been studies on people's attitudes towards automated driving, another important ITS concept—connected driving—has received little to no attention. In the current study, data on how connected driving is represented and perceived were collected from German(-speaking) Reddit and Twitter. In relevant Reddit entries, the necessity of communication between vehicles was discussed almost exclusively in the context of automated driving. On Twitter, mostly shared news and information on the topic are presented, while the number of personal opinions is low. The most concerning subtopic seems to be cybersecurity, which reflects a general trend of data protection issues discussed in society.

**Keywords:** ITS; intelligent transportation systems; connected driving; autonomous driving; social media; Twitter; Reddit

## 1. Introduction

### 1.1. Motivation

Automated vehicles, such as Tesla's Autopilot, are gradually becoming part of our reality (Honda 2020). An important keystone to accomplish the goal of fully automated driving is the implementation of connected driving or communication between the vehicles and other traffic participants, such as vehicle-to-vehicle (V2V), vehicle-to-infrastructure (V2I), and as a broader concept, vehicle-to-everything (V2X). Integration of telecommunications, electronics, and information technologies with transport engineering to provide innovative services relating to different modes of transport and traffic management are known as intelligent transport systems (ITS) (EU 2010).

Besides the obvious technological difficulties, there are other challenges related to connected and automated driving, such as legislation and public attitudes (Howard and Dai 2014). Concerns and opinions on AD have been widely researched (see Kyriakidis et al. 2015; Bakalos et al. 2020). However, connected driving has received far less attention.

In the last decades, social media platforms have become an important means of exchanging news and reactions to them, serving for both personal use and information dissemination by officials, news agencies, or experts. Opinion mining based on the analysis of social media data increases (Priya et al. 2019). Bakalos, Papadakis, and Litke (Bakalos et al. 2020) analyzed posts from Twitter and Reddit with the help of natural language processing (NLP) models to classify opinions on automated mobility and automated vehicles. Their research is integrated into a "Social media Periscope for You" (SPY) monitoring tool (ITML 2020). The study (and the tool) focuses on the texts in English, which is a dominating language in the field of NLP research (Mielke 2016). Furthermore, the geography of posts (and therefore, opinions) is poorly reflected because location information is not obligatory on Twitter and Reddit.

The current study aims to address the gaps identified above, namely to analyze social media content related to the topic of connected driving that is posted in German or by users from Germany. Thus, the research entails several directions not considered before and has an exploratory character. There are several research questions to be addressed, concerning German(-speaking) social media.

1. How much is written on the topic connected driving?
2. How often is connected driving mentioned on its own, not paired with automated driving, possibly in comparison to when it is mentioned together with it?
3. What are public opinions (if any) about connected driving?

### 1.2. Related Work

Connected and automated vehicle technologies are expected to be deployed in the near future. The acceptance of autonomous transportation is a well-studied topic (Becker and Axhausen 2017; Golbabaei et al. 2020; Nordhoff et al. 2019; Liu et al. 2019). Research on the public acceptance and perception of connected and automated vehicles suggests that several factors such as safety, costs, trust, or regulation influence public acceptance of the ITS (Shiwakoti et al. 2020). However, people's knowledge and opinions about the latest developments and implementations of connected and automated vehicles are the areas that are currently lacking sufficient investigation. Research has been undertaken on various aspects and applications of ITS technologies and infrastructures (Rana and Hossain 2021), but less attention has been paid to the opinions and personal experiences of people concerning ITS.

Social media data can be used as a valuable data source for analyzing people's opinions and personal involvement centered around certain events or topics, for example, political communication (Stieglitz and Dang-Xuan 2012; Tumasjan et al. 2010), social trends and activities (Mändli 2015), and transportation (Chaniotakis et al. 2016). Over the past few years, technologies of automated, connected, and smart mobility have become one of the hottest topics and are often discussed on social media, which is also reflected in the research. The SPY tool (ITML 2020) provides interactive visualizations of "aggregated social media data related to different topics on the shared, connected, electrified fleets of AVs in coordinated PT [public transport], DRT [demand-responsive transport], MaaS [mobility-as-a-service] and LaaS [logistics-as-a-service] operational chains". On the website, one can choose one of four categories (challenges, road safety, public transport, and logistics) and the period, from which the data should be analyzed. The visualizations include, among others, the top five most frequent words that appeared in the selected data, the most active users within the collected social media (Reddit and Twitter) entries, the associations between these two findings, and automatic sentiment analysis. The SPY tool collects data automatically based on a lexicon of English terms (see Bakalos et al. 2020). Such a principle of data collection does not allow the filtering of irrelevant entries, which is especially problematic in the case of abbreviations. Thus, DRT can be part of a Twitter username (@DrT_Crit_Think) and maaş, which is for Twitter query equivalent to MaaS, is a Turkish word meaning "salary" (Tureng Dictionary and Translation Ltd. 2022).

Research about connected driving already used social media for data collection (see Maghrebi et al. 2015; Grant-Muller et al. 2015; Salas et al. 2017), but the sources focused on research specific goals such as frequent activities and locations on a work day, filtering qualitative data for transport operators or policy makers, as well as figuring out if social media supports a real-time recognition of traffic incidents. Manual analysis of entries and social interactions can help to identify, classify, and understand currently popular and most discussed ITS-related topics (primarily connected driving) and opinions on them. This study encompasses relevant data collection and analysis from German(-speaking) Reddit and Twitter.

## 2. Materials and Methods

The two social media platforms, Reddit and Twitter, allow the scraping or automatic collection of publicly available data via application programming interfaces (APIs). Reddit serves as a news aggregation platform, where the entries (posts and related comments) are organized into thematized blocks, i.e., subreddits (for further qualitative differences between Reddit and Twitter see Priya et al. 2019). Twitter is a leading microblog platform, which is widespread in different population strata and age groups and can provide a representative sample for many research questions (Grillenberger 2021). Reddit imposes a 40,000 character limit for a post and 10,000 for a comment, while a Twitter text entry can be 280 characters only. Both platforms have a similar number of active users—ca. 430 million (see Bakalos et al. 2020; Statista 2022b) and are most popular in the USA, while only 3.34% of Reddit desktop traffic comes from Germany and there are 7.75 million Twitter users in Germany (Statista 2022a, 2022c). However, there are almost 100 times more tweets than Reddit entries produced every day (see Murphy 2019; Stricker 2014).

General procedures of data scraping and analysis are presented in Figure 1. The bigger steps are separated with the help of the color: technical requirements, data scraping steps, and data labelling.

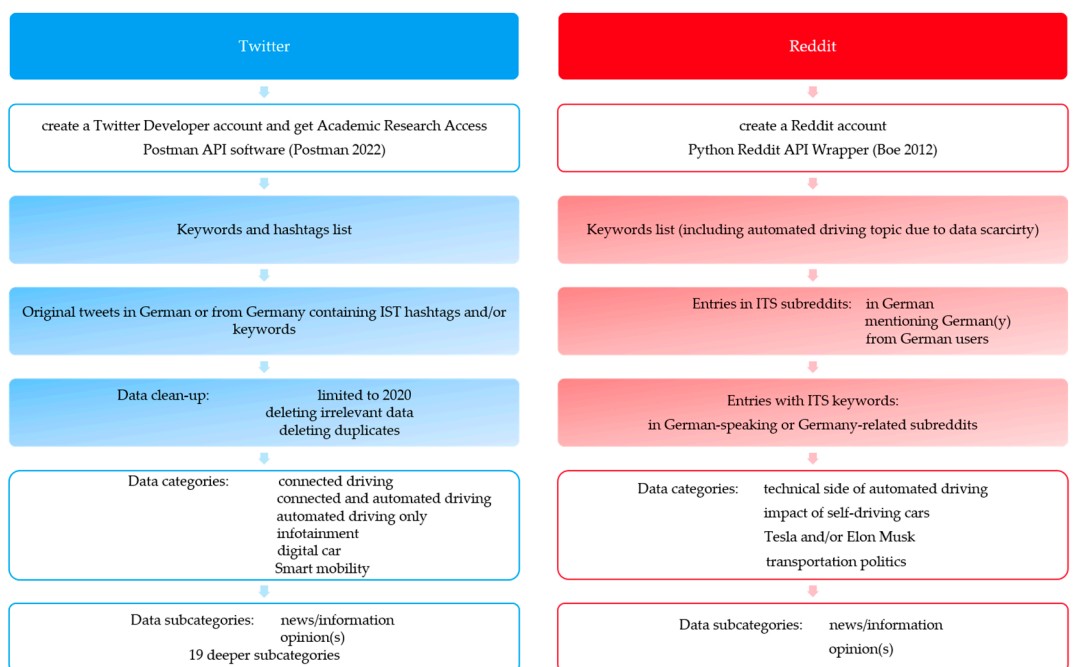

**Figure 1.** General procedures of data scraping and analysis (Postman 2022; Boe 2012).

The relatedness of text entries (tweets/posts and comments) to ITS topics was evaluated with the help of the keywords and/or hashtags (case can be ignored) displayed in Table 1 (cf. Bakalos et al. 2020).

A preliminary manual search showed that the amount of data on Reddit related exclusively to connected driving could be scarce. Therefore, it was decided to include keywords/hashtags from the topic automated driving into the search. Reddit entries can be relatively long, and it was assumed that a text on automated driving could include information on connected driving not necessarily in a way that would match our text search.

**Table 1.** Keywords and/or hashtags related to ITS topics.

| Reddit and Twitter | | |
|---|---|---|
| **Lang** | **Keywords** | **Hashtags** |
| DE | Vernetzung or Vernetzt* or Intelligent* followed by Fahren/Fahrzeug/Auto/Verkehr/Infrastruktur/Mobilität/Verkehrssystem (including declination forms, * stays for any possible grammatical ending) Verkehrstelematik | Verkehrstelematik, VernetzteMobilität, VernetztesFahren, vernetztesAuto |
| EN | C-ITS Internet of Vehicles | C_ITS, CITS, cooperativeITS, v2i, v2v, v2x, vehicletoeverything, vehicletoinfrastructure, vehicle2infrastructure, vehicletovehicle, vehicle2vehicle, C2C, C2X, CV2X, car2car, CarToCar, Car2X, CarToX, cartoinfrastructure, connectedcar, connectedcars, connecteddrive, connecteddriving, connectedmobility, connectedvehicle, connectedvehicles, intelligenttransportationsystem, intelligenttransportationsystems, intelligenttransport, intelligenttransportation, smartmobility |
| **Reddit only (keywords)** | | |
| DE | self-driving, driverless, robotaxi, robocar, autopilot, telematics autonomous/automated/connected/smart/intelligent followed by vehicle/car/bus/taxi/drive/driving/mobility/transport | |
| EN | selbstfahrend, Telematik, automatisiert/autonom followed by Fahren/Fahrzeug/Auto/Bus/Verkehr/Infrastruktur/Mobilität | |

### 2.1. Reddit

2.1.1. Reddit Scraping: General Procedure

As outlined in Figure 1, the search for relevant data was performed in several ways. Reddit scraping was performed through a number of subreddits with the help of Python package praw – Python Reddit API Wrapper (Boe 2012). The list of the ITS subreddits with the number of members in brackets as of July 2021 can be seen in Table A1 (Appendix A—cf. Bakalos et al. 2020). The subreddit *ConnectedVehicles* was not included in the search because it was not open to the public.

If an entry contained a text, the language of the text was detected with the help of the python library langdetect (Danilák 2021). If the first or the second most likely detected language was German, the entry would be selected for further analysis as "post/comment in German". No such entries were found. Additionally, a search with the regular expression "german" was performed. Forty-six posts/comments, in which Germany or relation to Germany is mentioned, were found.

To collect posts/comments made by active users (presumably) from Germany, a list of active users (presumably) from Germany was made first. "Active" means that these users had posted or commented in corresponding subreddits. Reddit API does not allow having a list of all subreddit members, but Reddit users can provide additional information about themselves using flairs. From subreddits *AskAGerman* and *germany* a number of user flairs were found, indicating where in Germany the user is from. If the author of an entry from subreddits *AskAGerman* and *germany* had one of these flairs, her name was added to the users_from_germany list, avoiding duplicates. Furthermore, the authors of entries from German regions and cities subreddits were added to the users_from_germany list, avoiding duplicates. The list of German regions and cities subreddits (the number of members in the parentheses) can be found in Appendix B. Users "user_id", "None", "converter-bot" were

deleted from the list because they either had been deleted or are not real people. The final list contained 48,504 users.

Next, entries from ITS subreddits were scraped. The entry was selected for further analysis if their author was in the users_from_germany list. From the content posted by some users, one could assume they were not from Germany (but usually from the US). If the activity of these users in other subreddits supported this version, their posts/comments were also deleted. The final collection of texts contained 240 posts and comments, 11 of which had duplicates among the mentions of Germany.

"German" subreddits, divided into several categories, are presented in Table A2 (Appendix C). If any of the ITS keywords/hashtags (see Table 1) were found in the post title or entry texts, the entry was selected for further analysis. The final collection of texts contained 73 posts and comments.

### 2.1.2. Reddit Data Analysis Method

The obtained texts were read and labelled according to four main topics, or data categories, and two further subcategories (see Figure 1). The opinions were further marked as positive or negative. Mentions of connected driving were marked separately. Data scarcity and heterogeneity did not allow applying a more structured and quantitative approach, but general perspective adopted the principles described in Section Quantitative Analysis.

### *2.2. Twitter*
### 2.2.1. Twitter Scraping: General Procedure

The first step of data scraping was performed with a full-archive search service in Postman (Postman 2022). Since the following analysis of data was manual, it was decided to limit the time of the search, namely starting in 2019 and ending in 2021.

Besides the keywords and hashtags in question, described above, the search query for Twitter data contained the following additional parameters:

- the entry must be either in German or if its location is known posted from Germany;
- the tweet is original, not a retweet;
- the tweet must be organic, not an advertisement.

Besides text information itself, the information on public metrics, place, from where the entry was posted (if given), and the author of the entry Initially, the data were retrieved for a limited amount of time, namely starting from January 2019, when the corresponding SaarMos-ITS project (FGVT 2019) began. There were more than 17,000 entries (tweets, referenced tweets, comments) in this sample. Due to a large amount of data, only the entries from 2020 (approx. 6000 items) were subject to further analysis. First, irrelevant entries to the ITS topic were removed. After this step, approximately 1300 items remained in the dataset. Further data reduction in favor of intensive analysis was based on the distinction of exactly identical tweets (duplicates), which remained only once per type in the data set. A total of 711 duplicates were identified within the 1300 entries. Thus, the final cleaned-up dataset consisted of 635 entries.

### 2.2.2. Twitter Data Analysis Method

In this research, a mixed methods approach[1] was used for both data collection and analysis and interpretation of the results. A classical approach, the convergent parallel mixed methods research design, was chosen, in which quantitative and qualitative methods are used in parallel to analyze data (Creswell and Creswell 2018). The quantitative analysis aimed to record and organize the most important topics in the area of "autonomous and connected driving" from the perspective of Twitter users in a category system. The primary purpose of the qualitative analysis was to take an in-depth look at Twitter users' communication on the topic of "autonomous and connected driving." Accordingly, relevant aspects were uncovered that could not be identified with the quantitative analysis.

Quantitative Analysis

The quantitative analysis closely follows Rössler's (Fraas et al. 2012) approach for quantitative online content analysis. Fraas et al. (2012) underline the following steps in Rössler's approach[2]:

iii. Development: formation of units of analysis with category systems

iv. Analysis: coding of data

In the current research, the formation of category systems (step iii) and coding (step iv) took place manually. The procedure represented a multi-step process in which categories were inductively[3] formed to gradually reduce the data until the central content could be recorded as a category. A subdivision into main, sub-category, and a deeper subcategory enabled a further structuring of the category system. In total, a category system with six main categories, two subcategories, and 19 deeper subcategories was developed throughout the analysis process.

*Main categories:*

1. CC (Connected Car)—connected driving in terms of V2X/V2V/V2I.
2. CC+AD (Connected Car and Automated Driving)—connected and automated driving.
3. AD (Automated Driving) ccHashtag (Connected Car Hashtag)—only entries related to automated driving; however, the hashtags correspond to connected driving— ccHashtag;
4. Infotainment—car radio, navigation system, hands-free phone, driver assistance systems, and other functions in a central control unit in the car, not referring to V2X;
5. Digital car—information about digitalization in the car (modern functions in the car, e.g., for better networking and safety)
6. Smart mobility—information about smart mobility topics, including connected driving.

*Subcategories:*

1. News/information
2. Opinion(s)

*Deeper Subcategories:* Cybersecurity, Events, Networking, Development of new systems to support connected and autonomous driving, Road safety, Studies, Infrastructure, New test track and fields, Mobility, Development of trial projects—and research ideas in institutions, New functions/equipment, AI, Reports, Ethics, Legal framework, Development of trial projects—and research ideas in companies, Overview of the current state of development of autonomous driving, Research, Importance of electronics and software.

Subsequently, the entries were transferred into six tables according to the main categories for an in-depth analysis. At first, the entries of each subset were classified into one of the two subcategories (news/information or opinion). Then, the entries were differentiated in their exact content (deeper subcategories). In this way, the entries could be better sorted thematically concerning the quantitative analysis.

In the final step, the results were summarized for each main category. Here, the entries were presented, sorted, quantified, and assigned anchor citations. Such summarizing tables served as a better representation of the results and the basis for the final analysis of the data.

Qualitative Analysis

The qualitative analysis was performed based on different utterances of Twitter users on the topic "autonomous and connected driving", which represented the subcategory "opinion". It was used to extract additional information from the data. For this purpose, the data of the subcategory "opinion" were analyzed qualitatively to find out which negative or positive topics related to autonomous and connected driving were discussed on Twitter[4].

## 3. Results

### 3.1. Reddit

From the relevant 46 entries, in which Germany/relation to Germany was mentioned, and 240 relevant entries made by active users (presumably) from Germany, most of the texts were found in subreddit *SelfDrivingCars* (40 and 215, respectively). From the 73 relevant entries found in German(y)-related subreddits, most of the texts (32) were from subreddit *de*. The earliest entries date back to 2013, the newest ones are from 2021. About 80% of the entries found are from 2018–2021. The full distribution of the collected entries according to the year of posting is shown in Figure 2.

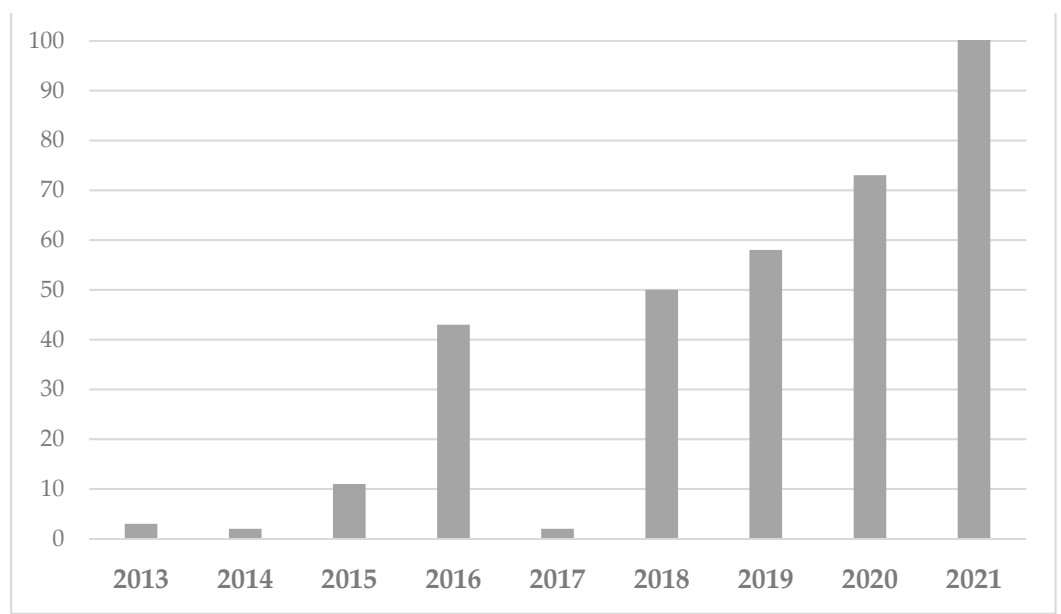

**Figure 2.** Distribution of relevant Reddit entries over time.

General analysis of Reddit data shows that there is little information related to connected driving. Relevant entries correspond mostly to the automated driving/self-driving cars topic. There are several major directions of discussion:

- automation will bring unemployment problems in general, self-driving cars—among truck drivers in particular;
- Tesla's autopilot in Germany (e.g., the question of its legal status);
- critique of Tesla and Elon Musk (rare pro-Musk entries);
- critique of German transportation politics/strong influence of the car industry, which leads to the fact that Germany is behind in the fields of electric and automated cars;
- doubts if higher levels of automation and self-driving cars will be available soon.

In total, there are six mentions of V2X communication necessity for automated driving (e.g., Figure 3) and one skeptical mention (see Figure 4), although there is more skepticism about 5G exclusivity for such purposes. Thus, connected driving is seldom mentioned or discussed on its own, only in the context of automated driving. There is one exception: an advertisement for the Mercedes-Benz E-400-4MATIC-Limousine with "Auto-to-X" option, which allows the exchange of information with similarly equipped cars to "see around the corner" and to recognize obstacles earlier (freely translated from German). This advertisement appears as a comment to a posted joke about Germany in subreddit *de*.

**Figure 3.** Statement about the necessity of V2X technologies for automated driving.

**Figure 4.** Skeptical opinion about the necessity of V2X technologies for automated driving.

*3.2. Twitter*

3.2.1. Presentation of Quantitative Results

As already explained, the subcategory "Opinion" is explicitly considered in the qualitative analysis, which means that the quantitative analysis focuses on the subcategory "News/Information".

Across the sample of 618 Tweets in total, Twitter users mostly shared news and information on topics related to connected cars (Connected Car—CC), automated driving (Automated Driving—AD), or a combination of both (see Figure 5).

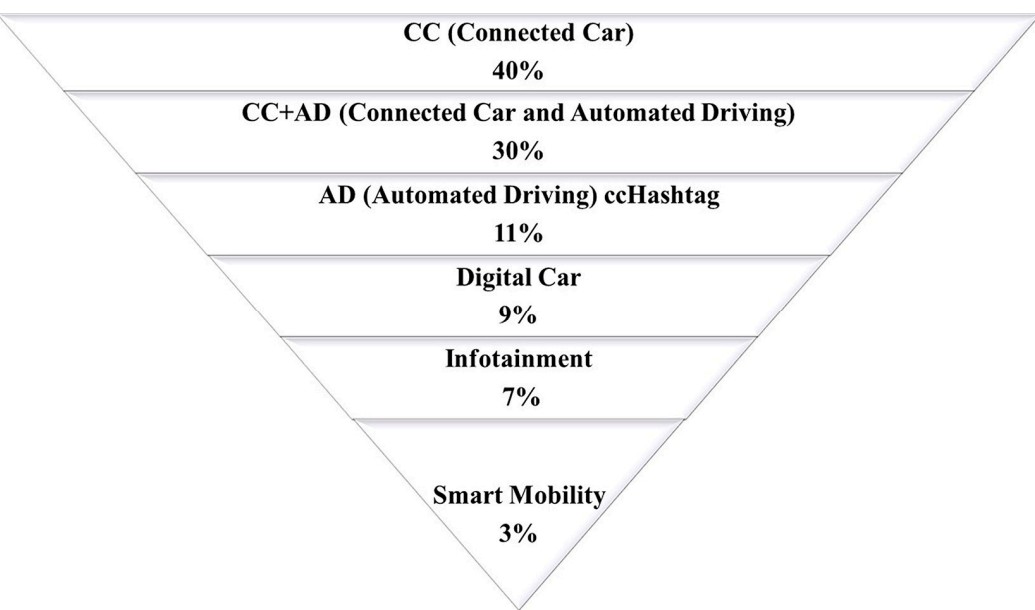

**Figure 5.** Topic distribution in the "News/Information" subcategory (n = 618 (100%)).

Since the first two categories are by far the most frequently mentioned topic areas, they are examined in more detail below.

The topic of "Connected car (CC)" considers connected driving in terms of V2X/V2V/V2I. Figure 6 demonstrates the topic distribution with some examples in the CC: News/Information subcategory[5]. Cybersecurity seems to be a highly relevant topic with almost 100 related tweets. It is followed by the topic of vehicle networking in general, with 20%. There are also discussions about events, developments of new systems to support connected and autonomous driving, road safety, various studies, and other content. Other content includes topics that were represented by fewer than six tweets.

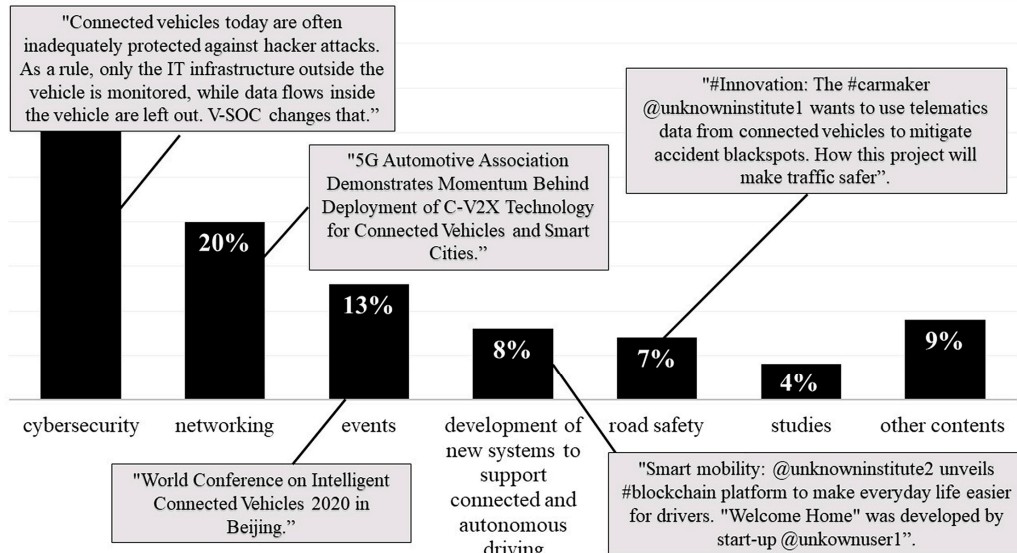

**Figure 6.** Topic distribution and example tweets in the category CC: News/Information.

Figure 7 shows the topic distribution and example tweets of the following level "CC + AD: Connected Car + Automated Driving". Entries related to connected and automated driving in combination are often about test tracks and fields. As in the previous category, topics such as networking and events share second and third place. Cybersecurity slips down a few notches with 12%. Additional topics include the development of experimental projects and research ideas in institutions as well as mobility. Road safety no longer seems to be a frequently mentioned topic. Selected examples with tweets are shown below in Figure 7.

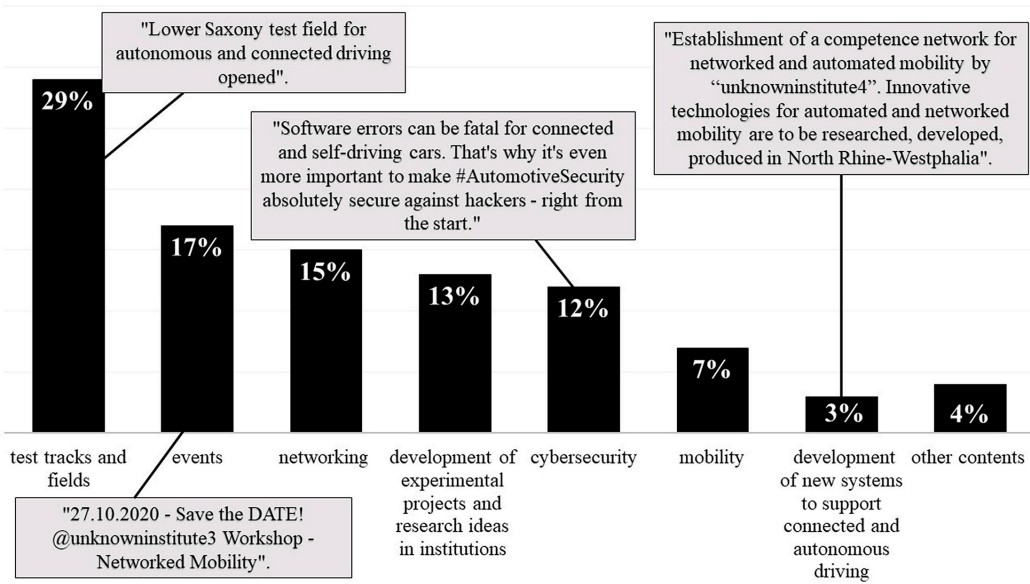

**Figure 7.** Topic distribution and example tweets in the category CC+AD: News/Information.

The tweets that contained a hashtag related to connected driving, but the content could be assigned to automated driving were classified under AD (automated driving) ccHashtag. The most discussed topic in this area was the development of new systems to support connected and autonomous driving—see the example in Figure 8. This was followed by aspects of networking in the area of 5G, AI technologies or information about research work in institutions.

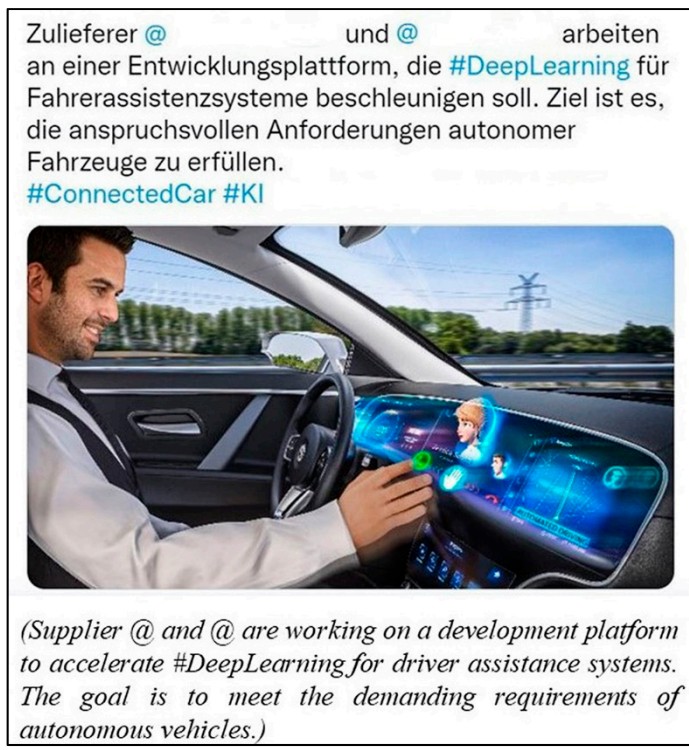

**Figure 8.** Statement about the development of new systems.

Digital cars as well as infotainment included 80% of tweets about the topic of new functions and equipment. In the last area, Smart Mobility, people mainly talked about upcoming events.

3.2.2. Presentation of Qualitative Results

A total of 16 opinion statements were collected, representing constructive statements made by users in the area of ITS. When adding the subcategory "Opinion" for the qualitative analysis, the distribution shown in Figure 5 does not change. Most of the opinion expressions are located in the main categories around connected and autonomous driving (CC, CC+AD, and AD ccHashtag).

Users are positive with their opinion that connected and autonomous driving increases safety—see the example in Figure 9.

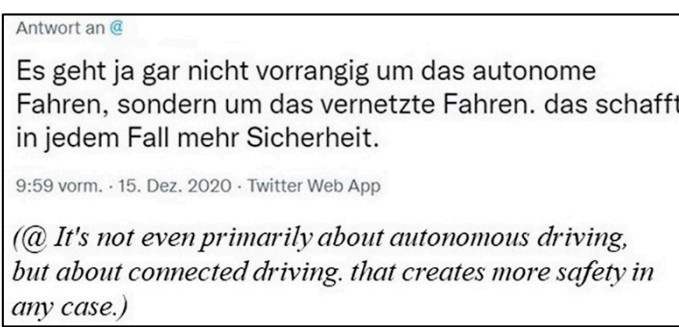

**Figure 9.** Statement about safety increase from autonomous and connected driving.

Likewise, the positive effects on the climate are addressed. A user therefore describes that connected and automated driving will deliver an important technology of the upcoming decades, which strengthens traffic safety and climate protection. Another user highlights the convenience that autonomous driving could bring and advocates for future autonomous mobility concepts—see Figure 10.

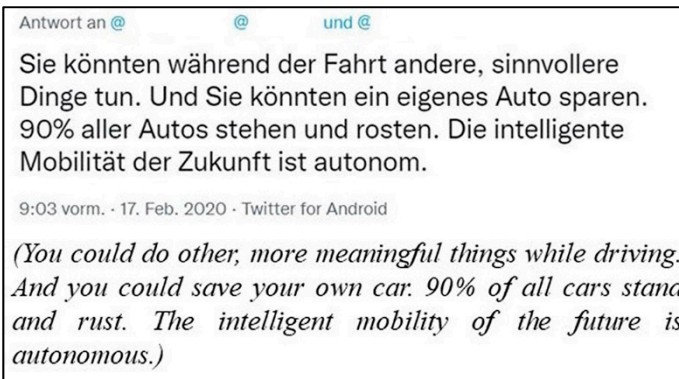

**Figure 10.** Statement about convenience from autonomous driving.

In the total number of filtered tweets under the subcategory of "Opinions," comparatively skeptical or negative statements regarding these topics dominate (eleven negative vs. six positive opinions). As emerged from the quantitative analysis, there is great relevance in topics such as cybersecurity. Many Twitter users report their fears and anxieties about surveillance and misuse of personal data by new technologies as can be seen in Figure 11.

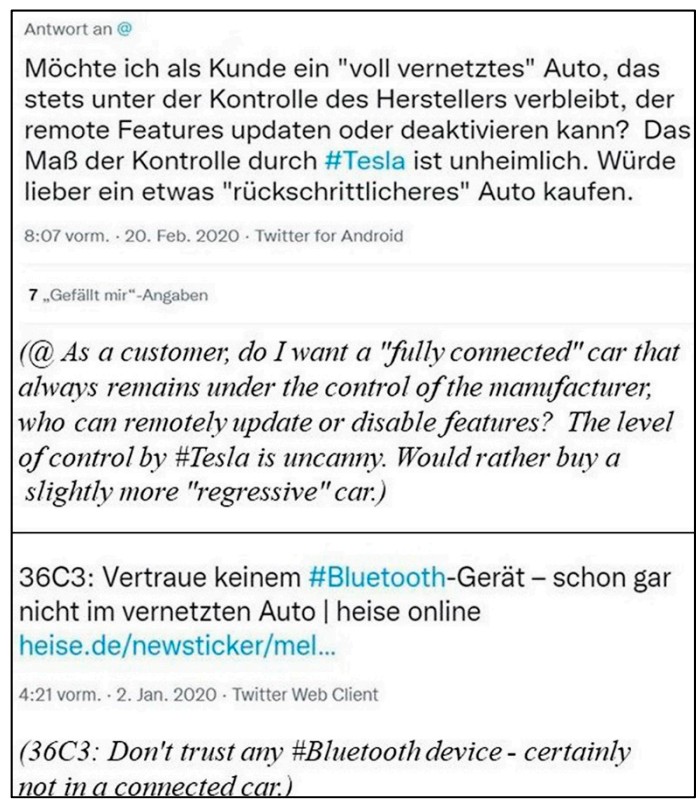

**Figure 11.** Skeptical and negative statements about personal data.

Autonomous driving in particular poses many dangers, according to Twitter users. For example, users do not believe in the perception abilities of autonomous vehicles. One can also notice that this negative opinion was supported by other Twitter users with likes and retweets. Furthermore, users criticize the decreasing control of the human driver due to many additional systems for connected and automated driving (see Figure 12). No or hardly any opinions could be assigned to the remaining main categories.

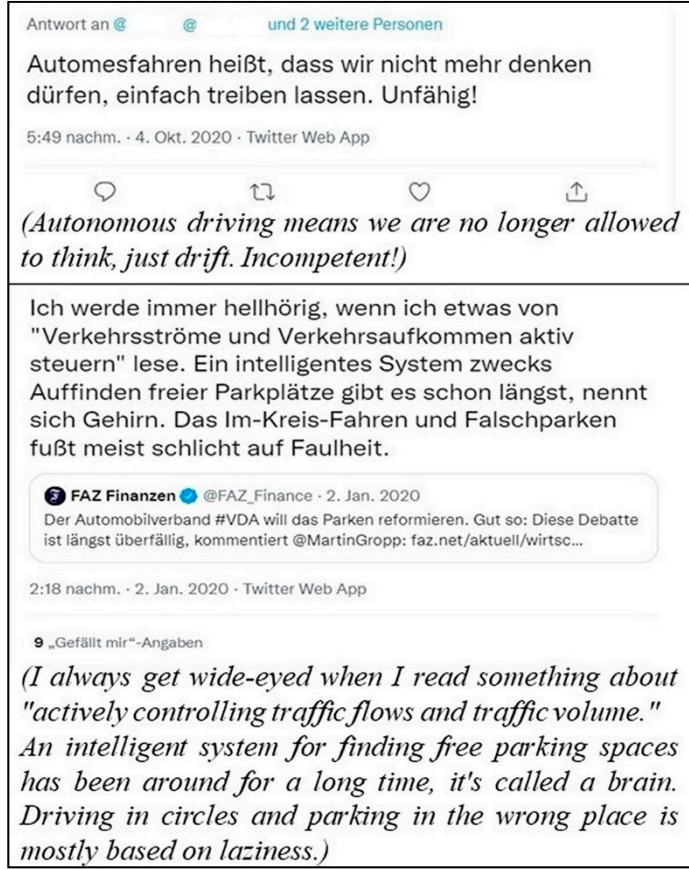

**Figure 12.** Skeptical and negative statements about connected and automated driving technology.

## 4. Discussion

The primary contribution of this study is the collection and classification of data on the topic "connected driving" in German(-speaking) Reddit and Twitter. Connected driving as a distinctive phenomenon had not been previously addressed with the material of social media. The collected and labelled data could be used in automated NLP applications for corresponding text classification, similar to the application built by Bakalos et al. (2020). On the other hand, the results of the study and the following suggestions can be used by the corresponding parties for public dissemination of information in the field of ITS and new mobility.

Reddit and Twitter scraping of entries related to connected driving in German(y) resulted in a large disproportion of data from these sources. Since almost 80% of the data collected from Twitter for 2020 had to be removed due to irrelevance, and the remaining data were halved because of the duplicates, the estimated amount of Twitter data in question for January 2019–July 2021 is more than five times greater than corresponding Reddit data, which had no time restriction but were, however, mostly from 2018–2021. This disproportion is in line with the findings of Bakalos and colleagues (Bakalos et al. 2020) on automated driving and reflects the general "massiveness" of Twitter in comparison to Reddit. However, there are important qualitative differences between the data found on Reddit and Twitter. First, only 25% of Reddit data was in German in comparison to monolingually German Twitter data. Furthermore, the data found on Reddit were almost exclusively dedicated to the topic of automated driving; the questions of connected driving, or more precisely, the necessity of communication between vehicles was discussed only in this context. This necessity is mostly recognized, although there is some skepticism, especially against 5G. The only independent commercial mention of V2X (which is the only mention in German) communication appears as a comment to an irrelevant post and receives no attention from other users.

Thus, one can conclude that connected driving is not represented in German-speaking Reddit. There is some discussion of the topic by the users (presumably) from Germany in specialized subreddits, predominantly in the biggest automated driving thematized subreddit *SelfDrivingCars*. This topic (and connected driving in its context) has steadily become more popular every year since 2018. A previous spike of entries in 2016 is probably caused by release of the first Tesla mass-market car.

In 40% of the analyzed dataset from Twitter, users share information about connected driving on its own. It is in balance with the automated driving context: when connected driving and automated driving are mentioned together (in tweet texts and associated hashtags), they account for about 41% of the data (see Figure 5).

The Twitter platform is used extensively for news dissemination. This is shown by the high number of duplicates that were found at the stage of data preparation. Among the topics in the field of connected driving, cybersecurity seems to be the most concerning one. This can be explained by people's general uncertainty about digitization and associated issues such as data protection. Such findings suggest that cybersecurity should be addressed not only by ITS researchers, but also in public dissemination of information.

The use of hashtags in the area of connected driving in tweets about automated driving shows that Twitter users might not yet be well versed in the distinction between "What is connected driving?" and "What is automated driving?". Many Twitter users are skeptical or negative about new mobility developments, which is also supported by the data from Reddit. That implies that social media users could benefit from more details on certain technology concepts, connected driving in particular. Better understanding of the concept and its distinction from automated driving—raising awareness, in other words—would help to foster public acceptance of connected driving.

Unfortunately, very few opinions could be filtered out and news and information prevailed in the data, which made it difficult to substantially address this question. Overall, however, it is striking that very little is said in German about ITS in general, considering the size of the Twitter platform. It is possible that the information is present "internationally" or in English (cf. Reddit results), which limits the awareness of those who acquire information "nationally". Encouraging corresponding parties to disseminate information in languages other than English would help to fill this gap.

One of the limitations of the study is its focus on Twitter and Reddit data, and the findings may not necessarily generalize to others who are not Twitter and Reddit users. Although some topics, such as political decisions regarding speed limits, can be understood and discussed by many users, other topics (e.g., the development of new systems to support connected and autonomous driving) are more likely to be very specific and are mainly discussed by users who already have certain experience in this area. In Twitter, the entries from both individuals and organizations were included in the analysis, which may limit the significance of the results.

This study does not claim to cover all possible topics related to ITS, while focusing on connected driving. Considering further topics may lead to a greater variety of categories. The frequency of mentions and the content of comments may also change over time (in this sense our study is only a snapshot), and some relationships between the categories formed may be identified, which has not been explored here. Future studies could validate or refine the coding, identify new categories, and explore changes in frequencies of mention and the content of comments, and the relationships between the categories. Online and offline surveys are necessary to extent the auditory—such results could be compared to the ones presented here and verify the representability of the latter. On the other hand, research on the material of other languages would complement the current study in creating a broader picture of "national" perceptions of the topic. This paper could be thus used as a starting point for further research and building a greater understanding of public awareness of ITS and attitudes towards its concepts.

**Author Contributions:** Conceptualization, E.R.; methodology, E.R., C.S. and O.D.; software, E.R.; formal analysis, E.R., C.S.; investigation, E.R.; resources, E.R.; data curation, E.R., C.S. and O.D.; writing—original draft preparation, E.R., C.S. and O.D.; writing—review and editing, E.R. and C.S.; visualization, E.R. and C.S.; supervision, E.R. and C.S.; project administration, H.W., E.R. and C.S.; funding acquisition, H.W. All authors have read and agreed to the published version of the manuscript.

**Funding:** The work was supported by the State Chancellery of Saarland during the project SaarMoS-ITS (grant number WT/2-SaarMoS-ITS). The consortium of the project consisted of the Saarland University of Applied Science (htw saar). All project results arise from the work of the SaarMoS-ITS project team of htw saar.

**Institutional Review Board Statement:** Not applicable.

**Informed Consent Statement:** Not applicable.

**Data Availability Statement:** Collected data are available upon request.

**Conflicts of Interest:** The authors declare no conflict of interest.

## Appendix A

**Table A1.** List of the ITS subreddits.

| Name of the Subreddit (as in Reddit) | Full Name/Description (Original Orthography and Punctuation Kept) | Number of Members |
|---|---|---|
| Futurology | a subreddit devoted to the field of Future(s) Studies and speculation about the development of humanity, technology, and civilization. | 15.4 mln |
| technology | Subreddit dedicated to the news and discussions about the creation and use of technology and its surrounding issues. | 10.6 mln |
| cars | the largest automotive enthusiast community on the Internet. We serve as Reddit's central hub for vehicle-related discussion including industry news, reviews, projects, videos, DIY guides, advice, stories, and more. | 2.4 mln |
| tech | The goal of /r/tech is to provide a space dedicated to the intelligent discussion of innovations and changes to technology in our ever changing world. We focus on high quality news articles about technology and informative and thought provoking self posts. | 383,000 |
| artificial | Artificial Intelligence | 141,000 |
| SelfDrivingCars | News and discussion about self-driving cars. | 66,900 |
| Automate | A place for the discussion of automation, additive manufacturing, robotics, AI, and all the other tools we've created to enable a global paradise free of menial labor. All can share in our achievements in a world where food is produced, water is purified, and housing is constructed by machines. | 40,400 |
| AutoNewspaper | Constantly Updated Automated News | 11,800 |
| SelfDrivingCarsLie | Self Driving cars are the biggest corporate lie today. They are not safe, they cannot save lives and they will fade away after the consumers will understand their limitations. The same people cheering for driving robots today will be the ones rejecting the technology tomorrow. Here is the proof to the lie. | 2500 |
| Driverless | Driverless Cars | 1500 |
| AutonomousNews | This subreddit is for the latest news happening in the world of autonomous vehicles and the underlying technologies that enable them. If there's an update to a 3D sensor, or a new start-up plans to build self driving cars, it belongs here. If Google announces their deep learning algorithms are getting even smarter or if there's news about the DARPA robotics challenge, post it. | 1200 |

**Table A1.** *Cont.*

| Name of the Subreddit (as in Reddit) | Full Name/Description (Original Orthography and Punctuation Kept) | Number of Members |
|---|---|---|
| AutonomousVehicles | focused on all autonomous technologies and news. No favoritism towards any brand, company, model, technology or otherwise. You may not agree on methods, but here we are focused on the tech involved. We will be inviting moderators from many different communities who have a relation to autonomous vehicles (not just cars), and all mod logs are public for the reason that this community is specifically intended to not show favoritism. | 1088 |
| transport | | 589 |
| SelfDrivingCarsTech | Self-driving cars technology news, articles, videos. Discussions on Localization, Mapping, Perception, Prediction, Planning, Control, and related questions | 422 |
| Autonomouscars | All About Autonomous Cars | 281 |
| V2X | Discussions and articles about Vehicle-to-Everything: communications with other vehicles, traffic lights, toll gates, pedestrians, homes and of course intra-vehicle gaming. V2V, V2I, V2X, V2P, V2H, 802.11p, MANETs, D2D | 252 |
| autonomousdriving | All About Autonomous Driving Vehicles | 248 |
| News_Automotive | Fresh and trending news about Automotive industry business, finance, employment, product, technology . . . and top companies in Automotive industry, posted by news bot power by AI. Welcome to join and share your ideas and opinions. | 94 |
| AutomotiveNews | | 83 |
| AutoDynamics | This is a subreddit dedicated to advancement in the automobile industry. | 69 |
| autonomous | shedding the legacy burdens of our past, moving towards a free, autonomous future | 66 |
| mobilityreport | description: Exploring the ideas & business of moving people. | 55 |
| ConnectedCars | A forum for sharing news, links and ideas for connected car apps and technologies. | 3 |

**Appendix B**

German regions and cities subreddits (the number of members in parentheses):

Bavaria (5.4k), Franken (454), Schwaben (445), Niderbayern (80), Niedersachsen (695), Brandenburg (246), SachsenAnhalt (156), Sachsen (217), Hessen (689), Thueringen (152), NRW (909), SchleswigHolstein (574), RheinlandPfalz (117), BadenWuerttemberg (1k), Meck-Pomm (82), Saarland (1.3k), Frisia (504); Berlin (119k), Hamburg (14.8k), Bremen (3.3k), Munich (45.8k), Nurnberg (11.7), Augsburg (5.9k), Regensburg (5.1k), Wuerzburg (5.1k), Erlangen (5k), Bayreuth (177), Bamberg (5.4k), Aschaffenburg (197), Landshut (113), Passau (1.7k), Freising (85), Straubing (1k), Erding (52), Kronach (15), Dorfen (22), Ingolstadt (280), jena (322), Stuttgart (24k), Heidelberg (16k), Mannheim (15k), Freiburg (5k), Karlsruhe (3k), Ulm (379), Kiel (1.2k), Norderstedt (65), Herrenberg (5k), Bielefeld (9.2k), Duesseldorf (4.2k), Dortmund (2.4k), Cologne (8.7k), Muenster (12.1k), Oberhausen (112), Dresden (7.2k), StadtEssen (639), Hannover (15.2k), Hildesheim (6.8k), Oldenburg (586), Leipzig (10k), Chemnitz (2.9k), Frankfurt (37.7k), Hagen (8.3k), Bonn (13.6k), Aachen (9.8k), Saarbruecken (161), Potsdam (344), Luebeck (200), Rostock (439), Tuebingen (1.5k), Esslingen (7.1k), Magdeburg (446), Wolfsburg (101), Osnabrueck (581), Koblenz (355), Kaiserslautern (604), Mainz (14.5k), Ilmenau (54), Erfurt (274).

## Appendix C

**Table A2.** "German" subreddits divided into categories.

| Category | Name of the Subreddit (as in Reddit) | Full Name/Description (Translated from German If Necessary) | Number of Members |
|---|---|---|---|
| General subreddits in German | de | The gathering place for all German speakers, mainly in German, sometimes also in English. For Germany, Austria, Switzerland, Liechtenstein, Luxembourg, and the two Belgians. | 456k |
| | mediathek | The best from the media libraries of the German-speaking world. | 830 |
| | Dokumentationen | A subreddit for documentaries in the German language. | 6.1k |
| | FragReddit | cf. /r/AskReddit (the place to ask and answer thought-provoking questions)—here in the German language version. | 32.7k |
| | Nachrichten | News for Germany. | 6.6k |
| | einfach_posten | just posting | |
| | GuteNachrichten | Good, positive news from Germany and the world. | 1.4k |
| | Deutschsprachig | For all German-speaking people! without censorship or great rules! Hate, spam, and co are of course not tolerated! Otherwise, you can post whatever you want to share with your fellow man! | 9 |
| Theme-specialised subreddits in German | Wissenschaft | News and interesting things from all areas of science. | 6.3k |
| | umwelt_de | German-language subreddit on the environment, nature, and environmental protection and technology | 2k |
| | digitalisierung | Digital- what? The gathering place for news, discussions, and questions about digitalization | 702 |
| | de_EDV | German subreddit for digital data processing: news, discussions, and help on hardware and software | 19.6k |
| | de_IT | IT in German | 2.7k |
| | Digitales | German news about the Internet, online, and technology | 100 |
| | technologie | News, questions, and discussions around the topic of technology. | 144 |
| | Verkehrswende | Germany—car country, that is still true. But a turnaround is in process, toward more cycling, public transport, and walking. | 32 |

**Table A2.** *Cont.*

| Category | Name of the Subreddit (as in Reddit) | Full Name/Description (Translated from German If Necessary) | Number of Members |
|---|---|---|---|
| | BerlinTransport | | 5 |
| Subreddits related to Germany in English | germany | English-language discussions and news relating to Germany and German culture. | 296k |
| | AskAGerman | Subreddit for all questions regarding Germany, Germans and life in Germany. | 21.2k |
| German regions and cities subreddits | | (see the list in Appendix B) | |

## Notes

[1]  "The central characteristic of mixed methods is the combination of quantitative and qualitative research steps in a study, which includes the equal collection and analysis of research data" (translated from German to English) (Grecu and Völcker 2017).

[2]  The first two steps (i. Analysis goal: Formulation of the analysis goal and ii. Planning of the analysis: definition of framework conditions, procedure) have already been performed and explained at the beginning of this paper.

[3]  Inductive means that the categories developed emerged from the data material. The aim was to achieve a "naturalistic" and "realistic" representation of the data material "without distortions due to the researcher's assumptions" (Mayring 2015).

[4]  "Qualitative results can contextualise quantitative results, providing a better understanding of the research problem because the results are more comprehensive, multi-perspective, and 'complete'" (translated from German to English) (Vogl 2017).

[5]  All exemplary twitter entries in this paper have been translated from German to English and all user and account names have been replaced by anonymous acronyms.

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
