# Peer review of "Connected Driving in German-Speaking Social Media"

_socsci, doi:10.3390/socsci12010046_

Round 1

Reviewer 1 Report

This paper aims to study people’s attitudes towards connected driving using German-speaking social media. In fact, study the attitude using social media is not new and there have been plenty of achievements. Anyway, it is still a hot research topic. The main problem is that there is a lack of methodology for carrying out this work. The method is very simple, i.e., collect data from Reddit and Twitter using existing tools, and the conclusion is also very simple. It is more like a technical report rather than a research paper. Some specific comments for the improvement:

1. The title is too big and should be modified. The title is "ITS in German-speaking Social Media", here ITS is too big, since this paper only considered connected driving.

2. More materials should be added related work, especially about connected driving research using the social media. Currently there is only about 2 papers are on this topic. Other are on social media or on ITS, though they have some relations with this paper, but the relation is very loose.

3. Section 2 should be compressed and most content can be deleted. Most of them are the details for the implementation. They are valuable to developing the software, but are not fit for a research paper.

4. One section about the method should be added. The method can be about the organization, the analysis, and the evaluation of the collected data. More attentions should be paid to how to process the data, not how to collect the data.

5. More valuable results should be explored, since current results are too simple.  

Author Response

Response to Reviewer 1 Comments

  1. The title is too big and should be modified. The title is "ITS in German-speaking Social Media", here ITS is too big, since this paper only considered connected driving.

Response 1: The title is modified to “Connected driving in German-speaking Social Media”.

  1. More materials should be added related work, especially about connected driving research using the social media. Currently there is only about 2 papers are on this topic. Other are on social media or on ITS, though they have some relations with this paper, but the relation is very loose.

Response 2: To the best of our knowledge, the presence of connected driving in Social media has not been studied. We have added, however, some works, in which the data from Social Media was used for the needs of connected driving, for example, for real-time recognition of traffic incidents

  1. Section 2 should be compressed and most content can be deleted. Most of them are the details for the implementation. They are valuable to developing the software, but are not fit for a research paper.

Response 3: Section 2 has been considerably reduced.

  1. One section about the method should be added. The method can be about the organization, the analysis, and the evaluation of the collected data. More attentions should be paid to how to process the data, not how to collect the data.

Response 4: Methods and Materials section has been reduced and better structured to make the presented work clearer.

  1. More valuable results should be explored, since current results are too simple. 

Response 5: The results are discussed in terms of their applicability for NLP applications, public dissemination of information on ITS, and further research extensions.

Reviewer 2 Report

1. The idea of Reddit and Twitter scraping entries related to connected driving in German is a very good innovation.

2. Also it is known Intelligent Transportation Systems (ITS) is one of the emerging areas that need to do research on this.

3. Clearly specify the contribution of the work in the paper.

4. The study is limited to only people who are connecting to Twitter and Reddit, and scope to extended work in future

5. The overall architecture of the study provided if possible

6.  Results need to write in a better  way

7. overall good

Author Response

Response to Reviewer 2 Comments

  1. The idea of Reddit and Twitter scraping entries related to connected driving in German is a very good innovation.
  2. Also it is known Intelligent Transportation Systems (ITS) is one of the emerging areas that need to do research on this.
  3. Clearly specify the contribution of the work in the paper.

Response 3: The results are discussed in terms of their applicability for NLP applications, public dissemination of information on ITS, and further research extensions.

  1. The study is limited to only people who are connecting to Twitter and Reddit, and scope to extended work in future
  2. The overall architecture of the study provided if possible

Response 5: A figure on data scraping and analysis is added.

  1. Results need to write in a better  way

Response 6: Changes have been made to the Results section to present the work in a clearer way.

  1. overall good

Reviewer 3 Report

The paper needs extensive proofreading. There are several issues with word choices and sentence structures that make it difficult to follow the text.

The title needs to be edited. ITS is a very broad term. The title must only refer to the investigated topic which is connected vehicles.

I disagree with the authors’ statement in the abstract which reads: “connected driving has received little to no attention”. There are many articles in this area. Perhaps the authors are referring to studies conducted in Germany. Clarification needed.

Authors bring up multiple research questions in section 1.1. They should get back to these questions in the discussion section and elaborate on their responses based on their effort in this paper.

The data collection procedure (explained in section 2) should be accompanied by flowcharts to better represent the procedure steps.  

Figure 1: Any discussion on the inconsistent pattern of data entries in 2017?

One of the main motivations of the study is to focus on public opinions about CVs in Germany. Are the results different from similar studies that are conducted in other geographical contexts?

While some of the discussions are interesting to me, I have a hard time understanding the big-picture idea here. What are the practical implications? Who will benefit from these findings and how? 

Author Response

Response to Reviewer 3 Comments

The paper needs extensive proofreading. There are several issues with word choices and sentence structures that make it difficult to follow the text.

Response 1: A native English speaker has proofread the text.

The title needs to be edited. ITS is a very broad term. The title must only refer to the investigated topic which is connected vehicles.

Response 2: The title is modified to “Connected driving in German-speaking Social Media”.

I disagree with the authors’ statement in the abstract which reads: “connected driving has received little to no attention”. There are many articles in this area. Perhaps the authors are referring to studies conducted in Germany. Clarification needed.

Response 3: To the best of our knowledge, the presence of connected driving in Social Media has not been studied.

Authors bring up multiple research questions in section 1.1. They should get back to these questions in the discussion section and elaborate on their responses based on their effort in this paper.

Response 4: The Discussion section has been revised and restructured according to the initial research questions plan.

The data collection procedure (explained in section 2) should be accompanied by flowcharts to better represent the procedure steps.  

Response 5: A corresponding Figure is added.

Figure 1: Any discussion on the inconsistent pattern of data entries in 2017?

Response 6: A previous spike of entries in 2016 is probably caused by release of the first Tesla mass-market car. (added to the Discussion)

One of the main motivations of the study is to focus on public opinions about CVs in Germany. Are the results different from similar studies that are conducted in other geographical contexts?

Response 7: To the best of our knowledge, there have been no studies focusing on this topic in other regions/languages.

While some of the discussions are interesting to me, I have a hard time understanding the big-picture idea here. What are the practical implications? Who will benefit from these findings and how? 

Response 8: The results are discussed in terms of their applicability for NLP applications, public dissemination of information on ITS, and further research extensions.

Round 2

Reviewer 1 Report

This paper has been improved much better.